# Moonlighting in *Bacillus subtilis*: The Small Proteins SR1P and SR7P Regulate the Moonlighting Activity of Glyceraldehyde 3-Phosphate Dehydrogenase A (GapA) and Enolase in RNA Degradation

**DOI:** 10.3390/microorganisms9051046

**Published:** 2021-05-12

**Authors:** Inam Ul Haq, Sabine Brantl

**Affiliations:** Friedrich-Schiller-Universität Jena, Matthias-Schleiden-Institut, AG Bakteriengenetik, D-07743 Jena, Germany; Inam.ul.Haq@uni-jena.de

**Keywords:** SR1P, SR7P, *B. subtilis*, GapA, enolase, degradosome, degradosome-like network, RNA degradation, small proteins, moonlighting activity

## Abstract

Moonlighting proteins are proteins with more than one function. During the past 25 years, they have been found to be rather widespread in bacteria. In *Bacillus subtilis*, moonlighting has been disclosed to occur via DNA, protein or RNA binding or protein phosphorylation. In addition, two metabolic enzymes, enolase and phosphofructokinase, were localized in the degradosome-like network (DLN) where they were thought to be scaffolding components. The DLN comprises the major endoribonuclease RNase Y, 3′-5′ exoribonuclease PnpA, endo/5′-3′ exoribonucleases J1/J2 and helicase CshA. We have ascertained that the metabolic enzyme GapA is an additional component of the DLN. In addition, we identified two small proteins that bind scaffolding components of the degradosome: SR1P encoded by the dual-function sRNA SR1 binds GapA, promotes the GapA-RNase J1 interaction and increases the RNase J1 activity. SR7P encoded by the dual-function antisense RNA SR7 binds to enolase thereby enhancing the enzymatic activity of enolase bound RNase Y. We discuss the role of small proteins in modulating the activity of two moonlighting proteins.

## 1. Introduction

Moonlighting proteins are proteins that exhibit more than one function (rev. in [1]). Some of the earliest evidence for protein moonlighting—before even the term “moonlighting” was coined—came from the analysis of group A streptococcal GAPDH (glyceraldehyde 3-phosphate dehydrogenase) in 1992. It was reported that a protein tightly adherent to the surface of the *Streptococcus pyogenes* M6 strain had sequence homology to other GAPDH proteins, and, surprisingly, did not only exhibit GAPDH enzymatic activity, but was also bound to lysozyme, cytoskeletal proteins and fibronectin [2]. Four years later, another group identified *S. pyogenes* GAPDH as a cell surface receptor for plasminogen [3].

In addition to abundant metabolic enzymes, chaperones display a number of moonlighting activities. A special case of moonlighting proteins are so-called trigger enzymes, metabolic enzymes that measure the amount of their metabolites and control—depending on metabolite availability—gene expression [4]. Trigger enzymes can exert their secondary function via binding to DNA, RNA, a protein or by protein phosphorylation. In the Gram-positive model organism *Bacillus subtilis*, examples have been found for each of these four moonlighting activities: the complex of the tRNA^Ile2^ lysidine synthetase TilS and the phosporibosyltransferase HprT binds DNA and activates *ftsH* transcription, aconitase CitB binds RNA and affects its stability, glucose permease PtsG phosphorylates GlcT, and glutamine synthetase GlnA binds transcription factors to modulate their activity (rev. in [4,5]). Moreover, the two metabolic enzymes phosphofructokinase (PfkA) and enolase (Eno) act in addition as scaffolding proteins in the *B. subtilis* degradosome-like network.

Recently, we have discovered two small proteins, SR1P and SR7P, that bind the glycolytic enzymes GapA (one of the two GAPDHs) and Eno, respectively. Whereas Eno was already known to act as scaffolding protein in the hypothetic *B. subtilis* degradosome [6], an additional activity in RNA degradation was not known before for GapA. We have shown that SR1P, which is expressed under gluconeogenic conditions [7] when the glycolytic activity of GapA is not required, binds GapA [8] to promote the GapA–RNase J1 interaction and to enhance the activity of GapA-bound RNase J1 in RNA degradation (rev. in [9,10]). It was unexpected that the second small protein discovered and further characterized by our group, SR7P, which is encoded by a dual-function antisense RNA, interacts with Eno [11]. We could show that SR7P binds to RNase Y-bound Eno and promotes degradation of two RNase Y substrates. As GAPDH is an abundant protein required for glycolysis in all organisms, it is not excluded that it moonlights in RNA degradation in other bacteria, too. Furthermore, we hypothesize that other bacterial species might also encode small proteins that impact RNA degradation by modulating degradosome components. 

In this review, we will first provide an overview of four types of moonlighting proteins found in *B. subtilis* and afterwards focus on GapA and Eno and the modulation of their moonlighting function by two small proteins. 

## 2. Moonlighting Proteins in *Bacillus Subtilis* Known until 2015

### 2.1. TilS-HprT—A Complex of Two Enzymes—Acts as Transcriptional Activator at the ftsH Promoter

The enzyme tRNA^Ile2^ lysidine synthetase TilS modifies cytidine in the first anticodon position of tRNA^Ile2^ to lysidine to allow recognition of the isoleucine codon AUA (rev. in [12]). HprT, hypoxanthine–guanine phosphoribosyltransferase is involved in the conversion of purine bases into nucleotides for purine salvage from degraded nucleotide acids [13]. Interestingly, the *tilS* and the *hprT* genes are located immediately upstream of the *B. subtilis ftsH* gene and overlap by one nucleotide. Surprisingly, a TilS–HprT complex acts as transcriptional activator at the promoter of the *ftsH* gene that codes for a cytoplasmic membrane protease important for membrane quality control and cell division [14]. Neither TilS nor HprT contain any known DNA binding motif, and there is no consensus between the two binding sites of the TilS/HprT complex, IR-I and IR-II, at the *ftsH* promoter. Apparently, the complex employs distinct and hitherto unknown binding motifs for its interaction with DNA (see Figure 1A). 

### 2.2. The Aconitase CitB Moonlights as RNA Binding Protein

Aconitase is a TCA (tricarboxylic acid) cycle enzyme that converts citrate into isocitrate, for which it needs a saturated iron–sulphur (FeS) cluster. When iron is scarce, the FeS cluster disassembles resulting in enzyme inactivation. The inactive enzyme has an alternative conformation and can now bind at iron-response elements (IREs) in the 5′ or 3′ UTR of mRNAs to repress translation or RNA degradation, respectively (rev. in [15]). In vitro, monomeric CitB binds at the IRE in the 3′ UTR of *qoxD* (cytochrome oxidase subunit), another IRE between the *feuA* and *feuB* iron uptake genes and a stem-loop in the 3′ UTR of *gerE* (sporulation transcription factor) mRNA [16] which might stabilize these RNAs, although no half-life measurements have been performed so far. By contrast, CitB binding to the 5′ UTR of *citZ* (citrate synthase) destabilizes the tricistronic transcript originating at the *citZ* promoter, thus preventing *citZ*, *icd* and *mdh* translation [17] (see Figure 1B).

### 2.3. The Glucose Permease PtsG Phosphorylates the RNA Binding Protein GlcT

*B. subtilis* transports glucose by the phosphotransferase system, the genes of which are encoded in the *ptsGHI* operon. PtsG, the glucose permease, can—in addition to its function in glucose uptake—also control the RNA binding activity of the antiterminator binding protein GlcT by phosphorylating or dephosphorylating it in response to glucose availability [18,19]. In the presence of glucose, the PTS takes up the sugar and transfers the phosphate group from PEP (2-phosphoenol pyruvate) via enzyme I (encoded by *ptsI*), Hpr (encoded by *ptsH*) and the EIIB domain of PtsG (encoded by *ptsG*) to the incoming glucose. At the same time, dephosphorylated GlcT dimerizes, binds and stabilizes an antiterminator stem-loop in *ptsGHI* mRNA to allow transcription of full-length *ptsGHI* mRNA. In the absence of glucose, PtsG phosphorylates GlcT instead of the sugar, thereby converting it to its monomeric form that is unable to bind and stabilize the antiterminator RNA. Hence, the 5′ UTR of the *ptsGHI* operon refolds resulting in formation of the more stable transcription terminator and, consequently, premature termination of *ptsGHI* transcription (see Figure 1C).

### 2.4. The Glutamine Synthetase GlnA Interacts with the Transcription Factor TnrA and Inactivates Its DNA Binding Activity

Under nitrogen limitation, *B. subtilis* glutamine synthetase GlnA catalyzes the ATP-dependent synthesis of L-glutamine from L-glutamate and ammonium while the pleiotropic transcription factor TnrA activates genes for the utilization of alternative nitrogen sources. In the presence of the preferred nitrogen source glutamine, GlnA is feedback-inhibited and forms a complex with either TnrA or GlnR, which abolishes the DNA binding activity of TnrA but stimulates that of GlnR. Whereas TnrA inhibits the enzymatic activity of GlnA, GlnR does not affect it [20]. The formation of the TnrA–GlnA complex depends on the presence of glutamine and AMP [21]. This provides a direct link between nitrogen supply, enzymatic activity of GlnA and TnrA-controlled gene expression (see Figure 1D).

### 2.5. Other Moonlighting Proteins 

In addition to the four types of moonlighting proteins described above, three enzymes act as scaffolding components in the *B. subtilis* degradosome-like network: Eno, PfkA [6] and GapA [9]. Until 2020, it was unclear if Eno or PfkA have an impact on RNA degradation or simply act as connectors for the other components. 

## 3. The *Bacillus Subtilis* Degradosome-Like Network (DLN)

### Composition of the B. Subtilis DLN

The degradosome is a multiprotein complex that is involved in the processing and degradation of bacterial RNA. It has been discovered in *E. coli* where its permanent components are the major endoribonuclease RNase E, the 3′-5′ exoribonuclease PNPase, the RNA helicase RhlB and the metabolic enzyme enolase (rev. in [22]). Later, degradosomes have been found in many other bacteria. Their composition is species-dependent and may also vary under certain growth or stress conditions. Recently, degradosomes of Gram-positive bacteria that—in contrast to the *E. coli* degradosome–could not be purified in the absence of cross-linking agents and seem to be more dynamic, were termed ‘degradosome-like networks’ (DLN) (rev. in [23]).

The *B. subtilis* DLN comprises the major endoribonuclease RNase Y [24], the 3′-5′ exoribonuclease PNPase [25], the exo/endoribonucleases J1 and J2 [26], the DEAD box helicase CshA [27], the metabolic enzymes Eno and phosphofructokinase PfkA [6,27] and under certain conditions glyceraldehyde 3-phosphate dehydrogenase A (GapA) [9] (see Figure 2). Using GFP fusions of all DLN components, only the membrane localization of RNase Y could be confirmed whereas no other protein was found exclusively at the membrane [28]. RNases J1 and J2 as well as CshA colocalized with the ribosomes, and, therefore, with the bulk of *B. subtilis* mRNA [28]. 

Using a bacterial two-hybrid system, nine interactions have been determined [6] which are shown in Figure 2B. Later, four of them could be corroborated by SPR (surface plasmon resonance), and Kd values for them were determined [29]: 5 nM (PNPase-RNase Y), 100 nM (Eno-RNase Y), 40 nM (Eno-PfkA), and 250 nM (PNPase-RNase J1), whereas an interaction between purified RNase Y and RNase J1 could not be confirmed in vitro. The measured four interactions have submicromolar Kd values suggesting that they are physiologically relevant. Proposed interactions between PfkA and either RNase Y or PnpA could not be confirmed by SPR, and CshA could not be studied because its domains were not soluble. 

Like many low GC Gram-positive bacteria, *B. subtilis* has two RNase J orthologs, J1 and J2 [26] which form a heterodimer that is most likely the predominant form in vivo [30]. RNase J2 is not essential and, later, it was shown that RNase J1 is also not essential, but a knockout causes severe defects in morphology, sporulation and competence [31]. Both RNases display a 5′-3′ exoribonuclease activity (although that of J2 is two orders of magnitude weaker) which is proposed to be their main in vivo activity, and an endoribonuclease activity, at least shown in vitro [30]. The association of RNases J1 and J2 has an effect on their endoribonucleolytic properties. While the individual enzymes have similar endoribonucleolytic activities and specificities in vitro, as a complex they behave synergistically to alter cleavage site preference and to increase cleavage efficiency at specific sites [30]. Structural studies on RNA-bound *Thermophilus* RNase J, which is similar to the *B. subtilis* enzyme, showed that the RNase J1/J2 dimer would need large reorganization to allow an endoribonucleolytic substrate to reach the catalytic site [32]. 

Based on the SPR results [29] and yeast two-hybrid and three-hybrid screens as well as a coimmunoprecipitation experiment with FLAG-tagged RNase J1 which did not detect RNase Y as interaction partner [30], it is unlikely that RNase J1 is a permanent component of the DLN. This is in line with its original purification in the ribosome fraction [26] and an almost simultaneously studied RNase J1-GFP fusion (formerly termed YkqC) where it colocalized with the ribosomes [33]. As mentioned above, this was confirmed in 2016 by another RNase J1-GFP fusion [28].

In *B. subtilis*, depletion of RNase J1 in a strain also lacking RNase J2 only modestly increased global mRNA stability from 2.6 to 3.6 min, and single mutants showed no effect [26]. This indicates that RNases J1/J2 are not the main endoribonucleases in *B. subtilis.* Nevertheless, in an RNase J1/J2 depletion strain, the levels of >650 transcripts were altered and the amount of more than 200 proteins was either significantly higher or lower [34].

For the major component of the *B. subtilis* DLN, RNase Y, it was demonstrated by end-enrichment RNA sequencing that a so-called Y-complex composed of the cytoplasmic proteins YlbF, YmcA and YaaT is involved in the RNase Y-dependent degradation of polycistronic mRNAs in *B. subtilis* [35]. By contrast, nearly no requirement of the Y-complex was found for the maturation of noncoding RNAs. The authors suggested this complex might be a specificity factor for RNase Y. We determined a three-fold increased half-life of the RNase Y substrate *rpsO* mRNA in an *ylbF* knockout compared to the wild-type strain [11] which supported these data. In 2021, an internal reflection fluorescence microscopy and single particle tracking study revealed that RNase Y diffuses rapidly at the inner membrane in dynamic short-lived foci that are not the active enzyme form because they are larger and more abundant after transcriptional arrest or in Y-complex mutations and do not depend on substrate RNA presence [36]. The authors observed that the Y-complex reduces number and size of the foci, i.e., the membrane assembly status of RNase Y, which allows the enzyme to sterically access and, consequently, cleave more efficiently bulky substrates like translated polycistronic mRNAs.

Depletion of RNase Y, the main endoribonuclease, increases the half-life of bulk mRNAs more than two-fold [24]. A transcriptomic study combined with Northern blotting revealed a global impact of RNase Y on mRNA stability in *B. subtilis* with 550 mRNAs accumulating through processing by RNase Y and 350 mRNAs [37] that were less abundant upon RNase Y depletion. 

## 4. GapA–SR1P

### 4.1. GapA, the Glycolytic Glyceraldehyde-3P-Dehydrogenase in Bacillus Subtilis, Can Bind Two RNases, RNase J1 and RNase Y and Moonlights in RNA Degradation

Glyceraldehyde 3-phosphate dehydrogenase (GAPDH) is an enzyme that converts glyceraldehyde 3-phosphate (G3P) into 1,3-bisphosphoglycerate, which is the sixth of nine steps in glycolysis (see Figure 3). The reaction catalyzed by GAPDH is the sum of two processes, the exergonic oxidation of glyceraldehyde 3-phosphate to 3-phosphoglycerate and the endergonic phosphorylation of 3-phospho-glycerate to 1,3-bisphosphoglycerate (dehydratization) using inorganic phosphate. This reaction is reversible. *Bacillus subtilis* has two GAPDHs, GapA and GapB, that can both catalyze the same reversible reaction. Whereas the NAD-dependent GapA is required for glycolysis, the NADPH-dependent GapB is necessary for gluconeogenesis [38]. The *gapA* and *gapB* genes are inversely regulated. 

GAPDH is a homotetrameric enzyme with a highly conserved structure. Thus far, only the structure of *Bacillus stearothermophilus* GapA has been solved [39], but it can be assumed that the structure of the *B. subtilis* GapA, whose monomer comprises 35.68 kDa, is very similar. GapA belongs to the 100 most abundant proteins in *B. subtilis* with 20,000 molecules per cell in glucose-minimal medium under logarithmic growth conditions [40]. 

Whereas GapB is degraded under glycolytic conditions when it is not required, GapA is stable under both glycolytic and gluconeogenic conditions [41]. This suggests that GapA might have an additional function under gluconeogenic conditions, when its enzymatic activity is not needed.

The bulk of GapA is localized in the cytoplasm [42], but the enzyme can be also loosely associated with the membrane in growing *B. subtilis* cells [43]. Interestingly, *Bacillus anthracis* GapA has been found even at the cell surface where it was able to bind plasminogen [44]. Other GAPDHs in pathogenic bacteria were also excreted by a so far elusive nonclassical pathway, e.g., in *Streptococcus pyogenes*, where GAPDH binds lysozyme and fibronectin [2] and group B streptococci, where it binds human proteins and stimulates B lymphocytes [45]. This demonstrates that glycolytic enzymes can moonlight in pathogenesis.

Furthermore, a link between replication and the three-carbon substrate part of glycolysis was found: in addition to deletions of *pgk, pgm* and *pykA*, a *gapA* deletion could suppress growth of at least three thermosensitive mutants encoding subunit DnaE of the lagging strand DNA polymerase III [46]. The authors hypothesized that the role of this link might be to modulate DNA synthesis in response to energy availability.

Surprisingly, in 2002 it was reported that GAPDHs of all three kingdoms of life can degrade RNA [47]. As representatives the authors analyzed rabbit, archaeal, *E. coli* and *B. stearothermophilus* GAPDHs. 

In 2010 we discovered that *Bacillus subtilis* GapA is bound by a small protein, SR1P [8]. In the course of our investigation of the SR1P-GapA interaction we found that purified *B. subtilis* GapA can neither bind RNA in a DRaCALA (Differential Radial Capillary Action of Ligand Assay) nor in an EMSA (electrophoretic mobility shift assay). In addition, it was not able to degrade RNA in an in vitro-RNA degradation assay [9] which makes it unlikely to be an RNase. However, as we could show in Far Western blotting and co-elution experiments, GapA can bind two *B. subtilis* RNases, J1 and Y [9]. Neither the GapA-RNase J1 nor the GapA-RNase Y interaction was bridged by RNA, i.e., they were direct protein–protein interactions. One of 80 GapA molecules contained RNase J1, and one of 100 contained RNase Y. This seems rather low, but one has to consider that GapA is at least 4- to 7-fold more abundant than RNase J1 (3000–5000 molecules/cell, [48]) and 10- to 100-fold more abundant than RNase Y (200–1400 molecules/cell, [48]) which would indicate that a significant amount of both RNases is bound by GapA. Northern blots with an established RNase J1 substrate, SR5 [49], revealed that the deletion of GapA affects SR5 degradation similarly as a deletion of RNase J1, namely 2.3-fold and two-fold, respectively [9]. This clearly demonstrated that GapA plays a role in RNA degradation. 

### 4.2. The Small Protein SR1P Impacts the Moonlighting Activity of GapA 

When we searched for new targets of the 205-nt long trans-encoded sRNA SR1 which is expressed under gluconeogenic conditions and repressed by CcpN and CcpA under glycolytic conditions [7], we found that SR1 is a dual-function sRNA: it acts as base-pairing sRNA in arginine catabolism [50,51], but is also an mRNA that encodes a 39 aa small protein, which we designated SR1P [8,10]. Interestingly, we observed high amounts of *gapA* mRNA in the presence of SR1P, and almost no *gapA* mRNA in its absence [8], although SR1/SR1P did not affect the half-life of *gapA* mRNA, i.e., SR1 did not directly stabilize *gapA* mRNA. Particularly surprising was that *gapA* mRNA was very abundant under gluconeogenic conditions, when its metabolic function was not needed which was later confirmed on the GapA protein level [41]. After we discovered that SR1P can interact with GapA, we focused on the elucidation of the biological role of this interaction. First, we applied DRaCALA, but neither GapA alone nor SR1P-GapA was able to bind RNA.

When we searched for further potential interaction partners of SR1P in co-elution experiments with protein crude extracts expressing Strep- or His_6_-tagged SR1P, we found—in addition to GapA as main interaction partner—RNases J1 and Y [9]. When we compared, in Far Western blotting and co-elution assays, GapA binding to the two RNases in the presence and absence of SR1P, we observed that SR1P enhanced the GapA-RNase J1 interaction about three-fold but did not affect the GapA-RNase Y interaction. We established an in vitro RNA degradation assay with two known RNase J1 substrates, SR5 [49] and threonyl tRNA [52]. Surprisingly, the trimeric SR1P–GapA–RNase J1 complex had a significantly higher RNase activity on both substrates than purified RNase J1 alone, whereas the GapA–SR1P complex isolated from an RNase J1 knockout strain was unable to cleave SR5. RNase J1 and the SR1P–GapA–RNase J1 complex yielded cleavage patterns of threonyl tRNA identical to that published before [52]. 

When we used Strep-tagged GapA to co-elute RNase J1 and SR1P and assayed the RNase J1 cleavage activity of preparations from logarithmic phase cultures (glycolytic conditions), we observed no RNase activity on SR5, whereas preparations from stationary phase cultures (gluconeogenic conditions) displayed RNase J1 activity [9]. This difference is due to repression of the *sr1* promoter under glycolytic conditions, yielding only 20 compared to 220–250 molecules of SR1/cell [51]. The growth-phase dependence of the RNase J1 activity of the SR1P–GapA–RNase J1 complex suggests that it is indeed SR1P that modulates the activity of GapA bound RNase J1. 

Our in vitro cleavage data could be substantiated in vivo by Northern blotting: The half-life of the RNase J1 substrate SR5 was two-fold higher in a Δ*rnjA* strain expressing *gapA* and *sr1* [49], 1.5-fold higher in a Δ*sr1* strain expressing *gapA* and *rnjA* and 2.3-fold higher in a Δ*gapA* strain expressing *sr1* and *rnjA* compared to the wild-type strain [9]. These data indicate that GapA moonlights in RNA degradation by RNase J1 under gluconeogenic conditions when its metabolic function is not required, and that SR1P, which is highly expressed under these conditions, modulates this moonlighting activity.

In 2012, we employed a bioinformatics approach that detected 23 SR1/SR1P homologues, exclusively in the order Bacillales, which showed an identical genomic location and high aa sequence conservation [53]. Nine of them were analyzed experimentally: except SR1P from *B. megaterium*, all could compensate a *B. subtilis sr1* deletion in vivo and the majority of them could interact with *B. subtilis* GapA revealing a remarkable conservation of SR1P over one billion years of evolution. 

To gain an insight into the GapA–SR1P interaction surface, we predicted the SR1P structure and confirmed it by CD measurements with chemically synthesized SR1P [54]. Using the published 3D structure of *B. stearothermophilus* GapA we searched and found an SR1P binding pocket: it comprises both the N-terminal helix 1 and the C-terminal helix 14 of GapA and contains three adjacent lysine residues, of which Lys 332 was decisive for SR1P binding [54].

We hypothesize that binding of SR1P to GapA might alter slightly the conformation of GapA to improve its interaction with RNase J1 which in turn might facilitate RNase J1 to bind and cleave its substrates. Recently, it was discovered that RNase J1 resolves stalled elongation complexes of the RNA polymerase [55]. It is tempting to speculate that GapA–SR1P may be also involved in this mechanism.

## 5. Eno–SR7P 

### 5.1. Enolase Moonlighting in B. Subtilis and Other Bacteria

Enolase is a metalloenzyme that catalyzes the reversible reaction of converting 2-phosphoglyceric acid (2PG) to phosphoenolpyruvic acid (PEP). The forward reaction occurs as the penultimate step during glycolysis while the reverse reaction takes place during gluconeogenesis. It requires certain divalent metal ions for its activity, among them Mg^2+^ as a naturally occurring co-factor [56]. Enolase was discovered in 1934 and is ubiquitously present in abundance in organisms capable of glycolysis or fermentation.

The *B. subtilis* Eno monomer has a mass of 46.58 kDa [57]. Using sucrose density gradient ultracentrifugation, the native mass of *B. subtilis* Eno was determined to be approximately 370 kDa [58]. In 2012, the structure of *B. subtilis* Eno was solved and showed that it forms an octamer [29].

Enolase is one of the most abundant proteins in many organisms, e.g., in *B. subtilis* there are ≈150,000 molecules per cell during exponential growth in glucose minimal medium [40]. In *B. subtilis* Eno is evenly distributed in the cytoplasm. In some cells it accumulates at the polar regions at 37 °C but not at 28 °C [28]. Polar localization of Eno is dependent on its phosphorylation on a tyrosine residue by the tyrosine kinase PtkA [59]. In the late stationary phase, Eno is also excreted as extracellular protein. A long, unbent hydrophobic α-helix comprising A108 to L126 is crucial for its extracellular secretion [60,61].

Like other glycolytic enzymes, Eno was considered an uninteresting and uncomplicated enzyme due to its high conservation over millions of years and straightforward role in glycolysis and gluconeogenesis. However, a number of recent studies have shown that besides its intrinsic role as a glycolytic enzyme, Eno plays an important role in several other biological and pathological cellular processes. These include a supportive role in the RNA degradation in *B. subtilis*, plasminogen binding and tissue invasion in *Streptococcus pneumoniae* and other pathogenic bacteria [62]. The moonlighting functions of Eno in a number of bacteria are discussed below.

In *B. subtilis* Eno interacts with two components of the DLN, the main endoribonuclease RNase Y and another glycolytic enzyme, PfkA [6]. Recently, we discovered another interaction partner for Eno, a small peptide SR7P that binds Eno and strengthens its binding to RNase Y. This in turn results in improving the RNase Y activity under certain stress conditions; (discussed in detail in 5.2 [11]). For RNase Y, the interaction with Eno plays a major role in the processing of its substrates. This was shown in vivo for one of the RNase Y substrates, *rpsO* mRNA. In an *eno* deletion strain, the half-life of *rpsO* mRNA was increased 2.5-fold [11]. Apart from its scaffolding function, another potential function for the interaction of Eno with PfkA still has to be studied.

In *Streptococcus pneumoniae*, enolase has two distinct moonlighting functions. In one case it acts as a plasminogen receptor [63], and in the second case it helps to evade the host immune system by binding the C4b-binding protein C4BP. This results in protection of the bacteria from complement-mediated killing [45]. Similarly, *Bacillus anthracis* Eno also binds plasminogen and laminin. In vitro, cells with surface Eno were capable of degrading fibronectin [64].

A recent study found that *Streptococcus thermophilus* enolase is capable of anchoring heterologous proteins to the *S. thermophilus* cell surface [65].

In addition to the examples discussed above, enolase has a moonlighting activity in a number of other pathogenic bacteria (rev. in [62]). Taken together, apart from its innate glycolytic activity enolase is involved in a diverse range of moonlighting functions in different bacteria and thereby enables these bacteria to properly respond to certain environmental conditions or to improve host cell invasion.

### 5.2. The Small Protein SR7P Modulates the Moonlighting Activity of Enolase

SR7P is a small protein (39 aa) from *B. subtilis* which is encoded by the dual-function antisense sRNA SR7 [11]. SR7 was previously published as S1136 by Mars et al. and reported to reduce the amount of the small ribosomal subunit under ethanol stress, most probably through transcriptional interference [66]. However, it escaped the authors’ attention that SR7 also has a small open reading frame (sORF). Our group has shown that this sORF is translated in *B. subtilis* and the peptide is produced under the control of a σ^B^-dependent promoter. The *sr7* gene is located in the intergenic region between *tyrS* and *rpsD*. SR7P is synthesized under ethanol, NaCl, manganese, heat shock and acid stress conditions from the σ^B^-dependent SR7 promoter as well as from a *tyrS* mRNA processing product that is produced constitutively under a σ^A^-dependent promoter.

SR7P was found to interact with Eno present in the *B. subtilis* DLN [11]. This interaction was confirmed by employing co-elution and Far-Western blotting. It was found that the interaction between SR7P and Eno in turn improves the binding of Eno to RNase Y. The interaction of Eno with PfkA, the other interaction partner of enolase in the DLN, was not affected by SR7P. The SR7P–Eno–RNase Y interaction occurs without bridging RNA [11]. 

The improved binding of enolase to RNase Y via SR7P modulates the activity of RNase Y (see Figure 3). The trimeric SR7P-Eno-RNase Y complex displayed a significant increase in the RNase Y activity compared to the binary Eno–RNase Y complex: RNase Y bound by Eno and SR7P showed in vitro a more efficient degradation of labelled *yitJ* 5ʹ UTR and *rpsO* mRNA than the binary Eno-RNase Y complex [11]. The role of SR7P in modulating the RNase Y activity via enolase was demonstrated also in vivo by measuring the half-life of *rpsO* mRNA. Both absence and overexpression of SR7P resulted in a small but significant alteration of the RNase Y activity.

By interacting with enolase-bound RNase Y, SR7P might act as sensor to link RNA turnover to different stress conditions, thus enabling *B. subtilis* cells to cope more efficiently with stress.

## 6. Conclusions

We discovered that one of the two GAPDHs in *B. subtilis*, GapA, is a moonlighting protein with a metabolic function in glycolysis and a second function in RNA degradation. As GAPDH is an abundant protein required for glycolysis in all organisms, it is not excluded that it moonlights in RNA degradation in other bacteria, too. 

The moonlighting function of *B. subtilis* GapA is modulated by the highly conserved small protein SR1P encoded by the dual-function sRNA SR1. Interestingly, SR1P seems to be confined to the order Bacillales, whereas GapA is a ubiquitous protein with conserved metabolic function and highly conserved 3D structure. Although we repeated our search in 2021 and could extend our list to 139 SR1/SR1P homologues (P. Müller, unpublished), none of them was found in other Gram-positive or Gram-negative bacteria that encode RNase J1/J2 or RNase Y homologues. Either we could not detect them because they are only structural and functional, but not amino acid sequence-homologues of SR1P, or GapA does not moonlight in RNA degradation in other RNase J1 encoding bacteria, and they employ instead other proteins to modulate their degradosomes or DLNs. Interestingly, *Staphylococcus aureus* was recently found to use the membrane protein flotillin that interacts with RNase Y as scaffolding protein in its DLN [67]. A third possibility could be that although in other Gram-positive bacteria, GapA is under certain conditions part of the dynamic DLN, it does not require a small protein to modulate its activity. Eventually, in pathogenic Gram-positive species like *S. pyogenes* or *S. aureus*, GapA might moonlight only in invader–host interactions [2,3], but not in RNA degradation. 

Since 1994, enolase has been known as scaffolding component of the *E. coli* degradosome (rev. in [68]), but it was also found in the *B. subtilis* [6], the *S. aureus* [69] and the *S. pyogenes* DLN’s [70]. The latter three species share not only RNases J1 and J2 (rev. in [70]) but also the endoribonuclease RNase Y as a component of the DLN although the deletion of *S. aureus* RNase Y affects only a small percentage of RNAs [71]. Whereas in *B. subtilis* and *S. aureus*, *rnjA* is not essential, its deletion is not viable in *S. pyogenes* [72]. We discovered that the moonlighting function of Eno is modified by the small protein SR7P [11]. Therefore, we hypothesize that *S. aureus* and *S. pyogenes* might also use small proteins as additional regulators of the moonlighting activity of Eno under certain environmental conditions. Similar to SR1P, our conservation analysis showed 10 SR7P homologues, only in the genus Bacillus, and comprising almost identical 20 N-terminal aa [11] which most probably constitute the SR7P/Eno interaction surface. Hence, it might be that in the case of Eno—as in the case of GapA/SR1P—either only structural and no amino acid sequence homologues of SR7P exist or that Eno in the other two species does not require a small protein for its interaction with RNase Y.

Currently, investigations are aimed at the detailed characterization of the GapA/SR1P and the Eno/SR7P interactions as well as on the identification of all RNase J1 substrates that depend on GapA/SR1P and all RNase Y substrates impacted by Eno/SR7P. Future research will reveal if the degradosomes or DLNs of other bacteria comprise GAPDH and if they are modulated by small proteins, too.

## Figures and Tables

**Figure 1 microorganisms-09-01046-f001:**
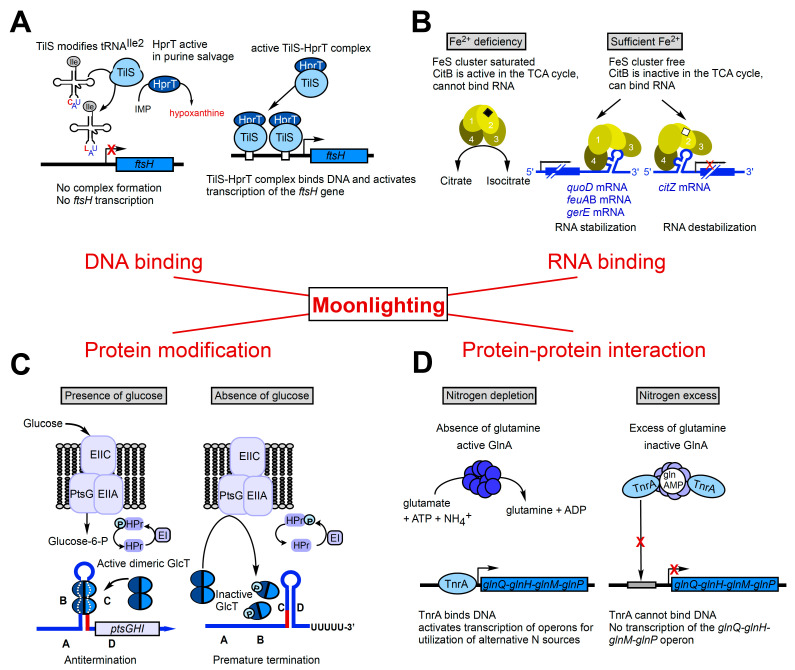
Examples of moonlighting proteins in *B. subtilis* that bind DNA, RNA or proteins or modify proteins. DNA is depicted as black lines, RNA as blue lines. Blue boxes, ORFs or operons; ovals or circles, proteins; arrows, transcription start sites; (**A**) the tRNA modifying enzyme TilS and phosphoribosyltransferase HprT can form a complex that binds DNA at two inverted repeats upstream of the *ftsH* promoter to activate transcription. (**B**) Under iron deplete conditions, CitB acts as aconitase in the TCA cycle, while under iron-rich conditions, it binds at the 5′ or 3′ UTR of mRNA to stabilize or destabilize it. Black diamond, FeS cluster. Numbers, CitB domains. (**C**) In the presence of glucose, glucose permease PtsG phosphorylates the incoming sugar while the dimeric GlcT stabilizes the antiterminator (B-C) in the 5′ UTR of *ptsGHI* mRNA thus ensuring transcription of the full-length *ptsGHI* mRNA. In the absence of glucose, PtsG phosphorylates GlcT preventing it from forming dimers and, consequently, RNA binding. This allows the 5′ UTR to fold into the more stable transcription terminator (C–D base-pairing) causing premature termination of operon transcription. (**D**) Under nitrogen depletion, GlnA synthesizes glutamine and TnrA binds DNA to activate transcription of operons for the utilization of alternative nitrogen sources. When nitrogen is in excess, the inactive, feedback inhibited GlnA binds and sequesters TnrA which thus cannot bind DNA to activate transcription of its operons anymore. For more details, see the text.

**Figure 2 microorganisms-09-01046-f002:**
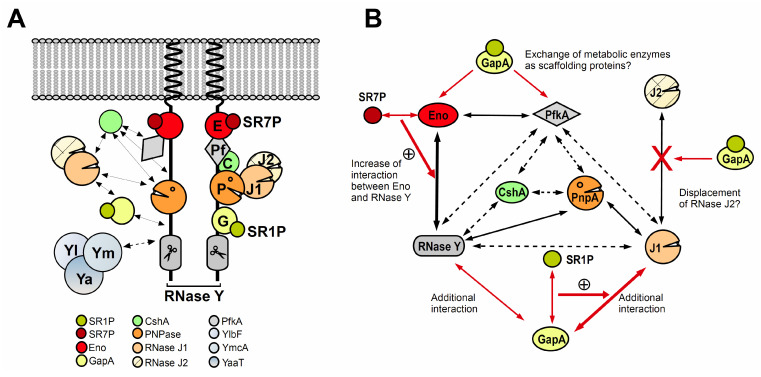
Interactions in the *Bacillus subtilis* DLN (degradosome-like network). (**A**) Hypothetical organization of the *B. subtilis* DLN considering all available experimental data [6,9,11,24,28,29,30]. Shown are the cytoplasm and the cell membrane in which the dimeric main endoribonuclease RNase Y (grey with scissors) is anchored. Its C-terminus provides a scaffold for the interaction with other DLN components: E, Eno; Pf, PfkA; P, PnpA; J1/J2, RNases J1 and J2; G, GapA. Small proteins SR1P and SR7P are depicted as solid circles. CshA (C) and the RNase J1/J2 heterodimer (J1/J2) are mainly located near the ribosomes (left side). However, based on the PnpA–J1 interaction, the latter might be occasionally associated with the DLN (right side). Yl, Ym, Ya symbolize the recently discovered Y-complex, the localization of which is still unclear (dashed arrow) but which is required for the degradation of mRNAs by RNase Y. (**B**) Working model for the role of SR1P and SR7P in the *B. subtilis* DLN. All interactions obtained using a bacterial two-hybrid system [6] are indicated by black arrows, among them dashed ones that were not corroborated by SPR [29]. Red arrows display interactions obtained by our work [9,11]. Thick red arrows, experimentally substantiated effects; +, increase in interaction and/or enzymatic activity; ?, hypothetical effects.

**Figure 3 microorganisms-09-01046-f003:**
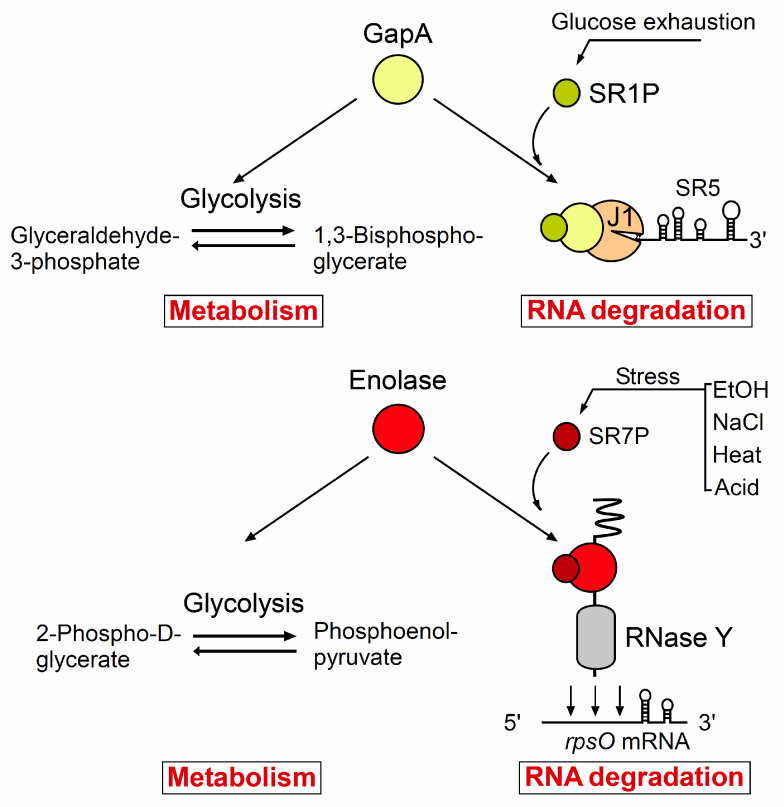
Modulation of the moonlighting function of the metabolic enzymes GapA and Eno by small proteins. Top, under glycolytic conditions, GapA acts as metabolic enzyme by converting glyceraldehyde 3-phosphate into 1,3-bisphosphoglycerate. Under gluconeogenic conditions, the small protein SR1P is expressed and binds GapA to enhance the GapA-RNase J1 interaction and to promote a more efficient cleavage of RNase J1 substrates, e.g., the RNA antitoxin SR5. Bottom, under glycolytic conditions, the metabolic enzyme Eno converts 2PG into PEP. Under diverse stress conditions (NaCl, ethanol, acid stress, heat shock) the synthesis of the small protein SR7P is increased; it binds Eno moonlighting in the DLN and enhances the enzymatic activity of Eno-bound RNase Y to cleave its substrates, e.g., *rpsO* mRNA.

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
