# Peer review of "Moonlighting in Bacillus subtilis: The Small Proteins SR1P and SR7P Regulate the Moonlighting Activity of Glyceraldehyde 3-Phosphate Dehydrogenase A (GapA) and Enolase in RNA Degradation"

_microorganisms, 2021, doi:10.3390/microorganisms9051046_

Round 1
Reviewer 1 Report
The review “Moonlighting in Bacillus subtilis….and Enolase in RNA degradation” by Haq and Brantl first describes examples of moonlighting proteins in B. subtilis and then focuses on two small proteins, SR1P and SR7P. SR1P and SR7P interact with GapA and Enolase, respectively, and affect their activity in the degradosome-like network. SRP1 strengthens the GapA-RNase J1 interaction; SRP7P stimulates the activity of RNase Y through its binding to Enolase. The review covers an interesting topic and is well written.
Comments:
1/ It will help with the readability of the Review if you explicitly state in the Introduction that you are first going to describe various types of moonlighting proteins in Bacillus subtilis and then describe in detail GapA and Enolase.
2/ Line 158 – RNase J1 is not essential (see EMBO J. 2020 Feb 3;39(3):e102500. doi: 10.15252/embj.2019102500.). Also, it may be worth mentioning/speculating that GapA-SRP1 may be perhaps involved in the ‘torpedo’ driven disassembly of stalled elongations complexes of RNA polymerase.
3/ Line 417 – Specify whether by “sequence homologues” you mean DNA or amino acids.
Author Response
1/ It will help with the readability of the Review if you explicitly state in the Introduction that you are first going to describe various types of moonlighting proteins in Bacillus subtilis and then describe in detail GapA and Enolase.
We have added the following sentence at the end of the Introduction:
In this review, we will first provide an overview of four types of moonlighting proteins found in B. subtilis and afterwards focus on GapA and Eno and the modulation of their moonlighting function by two small proteins.
2/ Line 158 – RNase J1 is not essential (see EMBO J. 2020 Feb 3;39(3):e102500. doi: 10.15252/embj.2019102500.).
We have added the reference Figaro et al., 2013 (new reference 31), where it was shown for the first time that RNase J1 is not essential:
RNase J2 is not essential, and later, it was shown that RNase J1 is also not essential, but a knockout causes severe defects in morphology, sporulation and competence [31].
Also, it may be worth mentioning/speculating that GapA-SRP1 may be perhaps involved in the ‘torpedo’ driven disassembly of stalled elongations complexes of RNA polymerase.
At the end of the paragraph on GapA/SR1P, we have added a hypothesis about the structural alteration of GapA and J1 induced by SR1P binding and, afterwards, introduced as new reference the torpedo paper (Sikova et al., new reference 55) which describes the torpedo function and the speculation about an involvement of GapA-SR1P this function of RNase J1 (new reference [55]
We hypothesize that binding of SR1P to GapA might alter slightly the conformation of GapA to improve its interaction with RNase J1 which in turn might facilitate RNase J1 to bind and cleave its substrates. Recently, it was discovered that RNase J1 resolves stalled elongation complexes of the RNA polymerase [55]. It is tempting to speculate that GapA-SR1P may be also involved in this mechanism.
3/ Line 417 – Specify whether by “sequence homologues” you mean DNA or amino acids.
Has been altered as follows:
Either we could not detect them because they are only structural and functional, but not amino acid sequence-homologues of SR1P, or GapA does not moonlight in RNA degradation in other RNase J1 encoding bacteria, and they employ instead other proteins to modulate their degradosomes or DLNs.
Reviewer 2 Report
Major:
Lines 50-62: The average reader may not be aware what the degradosome is, so a brief introduction to bacterial and B. subtilis RNA degradation would be welcome. Further confusing the issue: the word “degradosome” is used in the introduction, but in the abstract and section 3 the expression DLN is used. An introduction to DLN will also help the reader understand lines 139-145, where a comparison to the E. coli degradosome is made (a comparison that only makes sense once you know what the degradosome is).
Line 237: The authors state that “We will come back to this issue below”, but it is unclear where the discussion about RNase activity of GAPDH is located. Please indicate the relevant section.
Minor:
Line 4: In the title I suppose that it should be GapA (and not Gapa).
Line 32: It is not clear whether the detected enzymatic activity was glyceraldehyde 3-phosphate dehydrogenase activity.
Line 33: Add the word “was” after “but”.
Lines 41-42: The sentence could be improved to: “Examples have been found for each of these four moonlighting activities in the Gram-positive model organism Bacillus subtilis:”
Line 42: Please state the “day-job” of the TilS/HprT complex (i.e. when it is not moonlighting).
Lines 45-46: Sentence should read: “Moreover, the two metabolic enzymes phosphofructokinase (PfkA) and enolase (Eno) act in addition as scaffolding proteins in the B. subtilis degradosome-like network.
Line 71 and Figure 1A: The tilS and hprT open reading frames overlap by 1 nucleotide, but they are NOT fused in B. subtilis. See figure 1A in reference 14.
Line 92: “Black line” should be “black lines” and “blue line” should be “blue lines”.
Line 134: Please define the abbreviation DLN first time it is used in the main text (outside abstract).
Lines 143-144: RNase Y is localised at the membrane, it is not colocalised. Moreover, the localisation of several other DLN components have been examined by fluorescence microscopy. See for example Hunt et al 2006 (PMID: 17005971) and Cascante-Estepa et al 2016 (DOI: 10.3389/fmicb.2016.01492).
Line 156: Remove the word “As”.
Line 158: Reference 28 does not state that RNase J1 is essential in B. subtilis. It states that: “In B. subtilis, RNase J1 was originally thought to be essential until it was shown that knockout of the gene was possible”.
Line 170: If I understand correctly, then DLN is thought to be a network of dynamic/non-permanent interactions between proteins. Since there are no permanent components of DLN, I do not understand what is meant by “it is unlikely that RNase J1 is a permanent component of the DLN”. Is there are rule that a protein has to interact with RNase Y in order to be a permanent part of the DLN?
Line 178: Please clarify that the RNase J1/J2 mutant used in reference 31 is also a depletion of RNase J1 (and not a true knock-out mutation).
Line 236: It was not human GAPDH that we analysed in reference 44, it was rabbit GAPDH.
Line 237: It was not B. subtilis GAPDH that we analysed in reference 44, it was from B. stearothermophilus (see Figure 2C in reference 44).
Line 260: SR1P is introduced twice. First time on line 238 and again on line 260.
Line 313 and Figure 2A: Eno, Pfk, and PNPase are drawn as permanently in a complex with RNase Y, whereas CshA and the Y-complex are not. It is not clear what data is used for this choice.
Lines 352-352: It would be interesting to the reader to add a comment on whether the RNA decay effect of mutating eno would be seen for both active site mutations, truncated eno-versions and complete deletions (I was unable to find out which deletion was used in reference 11).
Line 369: Reference number for Mars et al. is missing.
Lines 428-429: Reference 68 contains the sentence “Deletion mutants of either RNase J exhibits very severe growth phenotypes and the correct maturation of essential molecules such as 16S rRNA and RNase P RNA are dependent on the RNase J1/J2 complex”, so I assume that RNase J1 and J2 are not essential in S. aureus (since deletion mutants can be obtained). Also, please provide a reference to state that RNase J1 and J2 from S. pyogenes are essential.
